# N-Beats as an EHG Signal Forecasting Method for Labour Prediction in Full Term Pregnancy

Thierry Rock Jossou [1,2], Zakaria Tahori [3], Godwin Houdji [4], Daton Medenou [2], Abdelali Lasfar [1], Fréjus Sanya [4], Mêtowanou Héribert Ahouandjinou [2], Silvio M. Pagliara [5,*], Muhammad Salman Haleem [5,*] and Aziz Et-Tahir [1]

1 Materials, Energy, Acoustics Team, Ecole Supérieure de Technologie de Salé, University Mohammed V in Rabat, Rabat 6203, Morocco
2 Department of Biomedical Engineering, Ecole Polytechnique d'Abomey-Calavi, University of Abomey-Calavi, Cotonou BP 2009, Benin
3 ENSAK, Universite Ibn Tofail Kenitra, Kenitra 14000, Morocco
4 Department of Computer Science and Telecommunication Engineering, Ecole Polytechnique d'Abomey-Calavi, University of Abomey-Calavi, Cotonou BP 2009, Benin
5 School of Engineering, University of Warwick, Coventry CV4 7AL, UK
* Correspondence: Silvio.Pagliara@warwick.ac.uk (S.M.P.); salman.haleem@warwick.ac.uk (M.S.H.)

**Abstract:** The early prediction of onset labour is critical for avoiding the risk of death due to pregnancy delay. Low-income countries often struggle to deliver timely service to pregnant women due to a lack of infrastructure and healthcare facilities, resulting in pregnancy complications and, eventually, death. In this regard, several artificial-intelligence-based methods have been proposed based on the detection of contractions using electrohysterogram (EHG) signals. However, the forecasting of pregnancy contractions based on real-time EHG signals is a challenging task. This study proposes a novel model based on neural basis expansion analysis for interpretable time series (N-BEATS) which predicts labour based on EHG forecasting and contraction classification over a given time horizon. The publicly available TPEHG database of Physiobank was exploited in order to train and test the model, where signals from full-term pregnant women and signals recorded after 26 weeks of gestation were collected. For these signals, the 30 most commonly used classification parameters in the literature were calculated, and principal component analysis (PCA) was utilized to select the 15 most representative parameters (all the domains combined). The results show that neural basis expansion analysis for interpretable time series (N-BEATS) forecasting can forecast EHG signals through training after few iterations. Similarly, the forecasting signal's duration is determined by the length of the recordings. We then deployed XG-Boost, which achieved the classification accuracy of 99 percent, outperforming the state-of-the-art approaches using a number of classification features greater than or equal to 15.

**Keywords:** N-BEATS; forecasting; labour prediction; electrohyterogram; deep learning

## 1. Introduction

This paper is a preliminary work investigating labour prediction in full-term pregnancy using a deep learning method to forecast EHG signals before the classification step, since most of the literature approaches are based on the EHG signals' classification through machine and deep learning methods alone. Although these methods provide good results, they are not yet suitable for real-time prediction and clinical applications. We therefore believe that the introduction of this signal forecasting step can take us one step closer to real-time birth prediction and clinical applications, since, in low-income countries, access to healthcare continues to be a luxury [1]. The situation is more complicated in rural and peri-urban areas, where the first and second delays of the three-delay model of emergency care are extended, and the risk of death is increased [2]. All this is due to the insufficient

number of healthcare centres in these areas and the lack of inadequate infrastructure allowing for the quick and efficient transfer of the expectant mothers. The early prediction of labour onset to ensure delivery in a health facility could provide an efficient way to reduce these delays. For this purpose, conventional clinical methods were less predictive in regard to pregnancy [3]. As an alternative, the electrohysterogram (EHG) [4], which represents the electrical signature of uterine muscle contractions, was found to be a good prospect for achieving this goal [5–8]. In this context, several studies, such as [9–11], have been carried out based on the external acquisition of EHGs from the pregnant woman's abdomen. The electrodes used and the different configurations proposed by most of these works were summarised in [12]. This enabled the extraction of several parameters characterising this technique, which can then be used to distinguish between contractions and non-contractions and to identify those that will lead to labour [13]. This equipment has been also used to characterize and classify preterm and term EHG signals [14].

Several classification methods have been developed and applied in order to distinguish between contractions and non-contractions based on EHG signals. The very first of these were statistical methods, which had limitations in terms of their ability to acquire a full-feature set representing contractions. Machine learning techniques were then used [15,16], which offered convincing results. However, traditional machine learning is usually computationally inefficient when applied to real-time signals and requires feature selection methods as a prior step to identify the useful features for classification, resulting in the loss of information. In recent years, the application of deep learning methods [16,17] to EHG signals has confirmed the importance of artificial intelligence in predicting labour. However, to the best of our knowledge, no research has been implemented in the clinical setup. Recent work has demonstrated the multifractality of EHG signals and that this characteristic can be exploited to monitor the progress of a preterm or full-term pregnancy [18,19].

The clinical applicability of the research work requires the involvement of real-time signals in the delivery prediction procedure. This includes the forecasting of time-series-based EHG signals before the classification techniques' application. Several statistical procedures have been deployed in order to forecast time series signals; however, they were unable to capture the non-linearity, randomness, and unpredictable tendency of these signals due to a lack of temporal context information [20–22]. In this regard, deep learning methods have been developed due to their ability to capture real-time features based on the temporal context [23]. According to the literature, although the EHG signal is a time series, no study has addressed its forecasting to date. This seems to be an important step in labour onset predictions in real time. Thus, labour prediction has remained at the stage of uterine contraction classification. In the literature, a number of studies used 15 parameters [24] and more [25] for the purpose of labour prediction, and these are often chosen without clearly established criteria. Meanwhile, as demonstrated in [26], an EHG signal's characteristics depend on anthropometrics and pregnancy variables. For all these reasons, our model automatically selects the 15 best features from the 30 that are most frequently used in the literature using principal component analysis (PCA). This technique was applied to the forecasted signal obtained using the neural basis expansion analysis for interpretable time series (N-BEATS) method [27]. Finally, XG-Boost was chosen as the classification algorithm according to its performance compared to Support Vector Machine (SVM) and other classification algorithms.

The structure of the paper is organised as follows: Section 2 presents previous works that have applied machine and deep learning methods to EHG signals. This is followed by Section 3, which details the data and methods used in the present work. Section 4 presents the results that are discussed in Sections 5 and 6, which concludes this paper.

## 2. Previous Work Using Machine and Deep Learning

It was found in the literature that several studies have applied machine and deep learning methods to EHG signals to make predictions through classification. We noted that there were more applications of machine learning methods than deep learning in these

studies [15,16]. Most of these studies use public databases of physiological signals, especially the Physiobank TPEHG [15,25,28–36]. Three recording channels (channel1 (E2-E1), channel2 (E2-E3), and channel3 (E4-E3)) based on four physiological electrodes were used to create this database. It is the only public database with a minimum number of electrodes (four electrodes) for the bipolar recording of EHG signals with satisfactory results. It fills the critical gap in the field data, especially in the context of low-income countries. Considering the continuous nature of the signals, one of the most common methods used to extract relevant information is empirical mode decomposition (EMD) [15,35], which decomposes the signal into different components to extract the features of the signals (frequency spectrum, cycle, etc.). Additionally, some of the studies used wavelet decomposition to analyse the transient behaviour of the signals [37]. One study used five public databases consisting of recordings of EHG signals that were combined or not combined with other physiological signals (foetal heart rate, cardiotocography, foetal ECG) [16]. Very few studies obtained results using their own experimental data [38–42].

Among the machine learning techniques for predicting labour, the Support Vector Machine seems to be the most frequently used and best performing machine learning method due to its ability to classify discrete features based on hyperplanes [15,28,31,33,36,38,41]. There have been few attempts to deploy artificial neural network (ANN) methods [16,33,34,42], and with the advancements in advanced artificial-intelligence-based methods, deep learning methods have also been deployed. Among the deep-learning-based methods, convolutional neural networks (CNN) [16,17] have been deployed to extract static features. However, considering the discrete nature of the training datasets, the strength of the deep learning methods has not been fully explored, resulting in an underperformance compared to that of traditional machine learning methods [16].

According to the literature, no study has investigated the prediction of labour based on EHG signal forecasting. However, we believed that it would be useful to introduce EHG signal forecasting over a given time horizon into the prediction model. This should be undertaken as a crucial step before the application of classification methods. It would render the prediction of labour onset in real time possible. Since the EHG is a time series, time series forecasting methods could be applied to it. As statistical methods have failed because they cannot capture the non-linear trend of these signals, deep learning methods are increasingly used today. These forecasting techniques have been used for several signals in different areas, with interesting results. For example, recently, these methods have been used for crude oil time series forecasting and COVID-19 disease recognition [21,23].

Most of the existing methods make predictions (especially those of preterm births) by classifying them on the basis of EHG signals and other physiological signals of the pregnancy or foetus. In contrast to the existing methods, our aim was to predict labour on the basis of the detection of continuous contraction signals. From this perspective, we propose the idea of a real-time pregnancy monitoring system which can predict labour based on real-time contractions. The real-time contractions can be detected via EHG signal forecasting. Therefore, we applied continuous EHG signals from the available public datasets in order to achieve our goal.

## 3. Used Dataset

The EHG signals used in this work were obtained from the publicly available Term-Preterm EHG Database on Physiobank [43]. Three recording channels (channel1 (E2-E1), channel2 (E2-E3), and channel3 (E4-E3)) based on four physiological electrodes were used to create this database. In order to remain consistent with the literature regarding the electrode positions providing the most usable EHG signals [12], we opted for signals from channel3. As we were interested in full-term pregnancies, we selected EHG signals recorded after 26 weeks of gestation for pregnancies that ended in full-term delivery. Among these signals, those that were filtered with a bandwidth of 0.3 to 3 Hz were considered to take into account the elimination of cardiac and respiratory rhythms [29]. Each signal obtained comprised 36,000 values recorded over 30 min.

## 4. Materials and Methods

The block diagram (Figure 1) below shows the different steps of the methodology that we followed in this study. It consists of the blocks of pre-processing, signal forecasting on a given horizon, the extraction and selection of parameters, classification, and decision. Each of these blocks will be described in the following subsections.

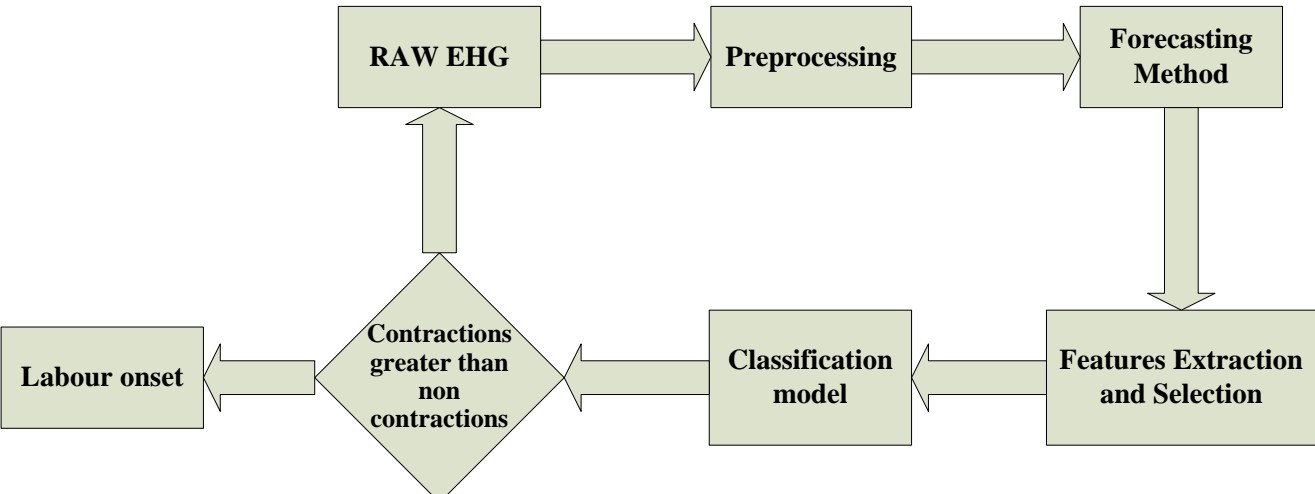

**Figure 1.** Methodology block diagram.

### 4.1. Dataset Pre-Processing

The empirical mode decomposition (EMD) method was applied to eliminate unnecessary variations in the signals and obtain the intrinsic mode functions (IMF) [44] more representative of contractions and non-contractions. The second IMF was considered in our case, as it is more representative, as confirmed in [36].

### 4.2. Feature Extraction and Selection

In our study, we considered the thirty most commonly used parameters (see list in Appendix A) in the literature to characterize uterine contractions. The fifteen best parameters for uterine contraction characterization were extracted from these using the principal component analysis method. The selected parameters were among those most used in the literature for EHG signal processing. The principal component analysis (PCA) method, which is a widely used dimension reduction technique [45] that does not result in the loss of essential information [33], was applied to the 30 parameters calculated previously. This provided us with the principal components, from which the most representative of the important characteristics of the signal were retained. After the analysis, the first 15 principal components were found to have the greatest variance and, therefore, to best represent the EHG signals under investigation. Here, the usefulness of the PCA lies in the fact that the representative features of the principal components depend on the anthropometric variables and the pregnancy and, therefore, are not the same from one pregnant woman to another. It is therefore a necessary step that must be undertaken before classification, since we can be sure that the results it provides will be the best in each case in terms of the representativeness of the signal information. These 15 principal components were therefore used as the input parameters for the classification model.

### 4.3. The Deep Learning Method for EHG Signal Forecasting

The stationarity of the data series to be sent to the forecasting algorithm must be ensured. For this purpose, the method of the visual observation of the signal, its mean, and its standard deviation, which are all represented on the same graph, is used. A statistical method is combined with this method for increasingly complex data series. In our case, the augmented Dickey–Fuller test [46] was used, as it is the most widely used statistical

method of stationarity analysis in the literature. It is a mathematical method based on a null hypothesis and an alternative one. The non-stationarity of the data is often considered as a null hypothesis. The objective is to eliminate it by obtaining a *p*-value lower than 0.05.

The forecasting model that we used in the present work is a model that was recently developed and dedicated to time series forecasting [27]. It is an architecture that is built on the advantages of the long short-term memory (LSTM) architecture. As the literature shows, LSTM neural networks were developed mainly to address the vanishing gradient problem [20,22], which other recurrent neural network (RNN) models were unable to solve. The neural basis expansion analysis for interpretable time series (N-BEATS) is an existing method that was proposed by [27] and a model which reinforces this property of LSTM networks and allows for the better capture of time series' non-linearity through its very deep architecture.

This architecture is a set of successive stacks of blocks, and each block is a fully connected multilayer network. When considering any block, it receives an input and provides two outputs: one corresponding to the best estimate of the input, and the other corresponding to the forecast of the block over a given horizon. Except for the first block, which has a global input, the other blocks receive the previous block' input estimation error as an input. The process continues in this way with regard to the inputs and outputs at the level of the stacks. N-BEATS is a pure, interpretable deep neural network whose architecture is based on backward and forward residual links (see Figure 2). This model has been shown to outperform the hybrid model that won the M4 forecasting competition by 3% [27]. For its implementation, PyTorch Forecasting was used, itself being a Pytorch-based package based on an open-source machine learning framework.

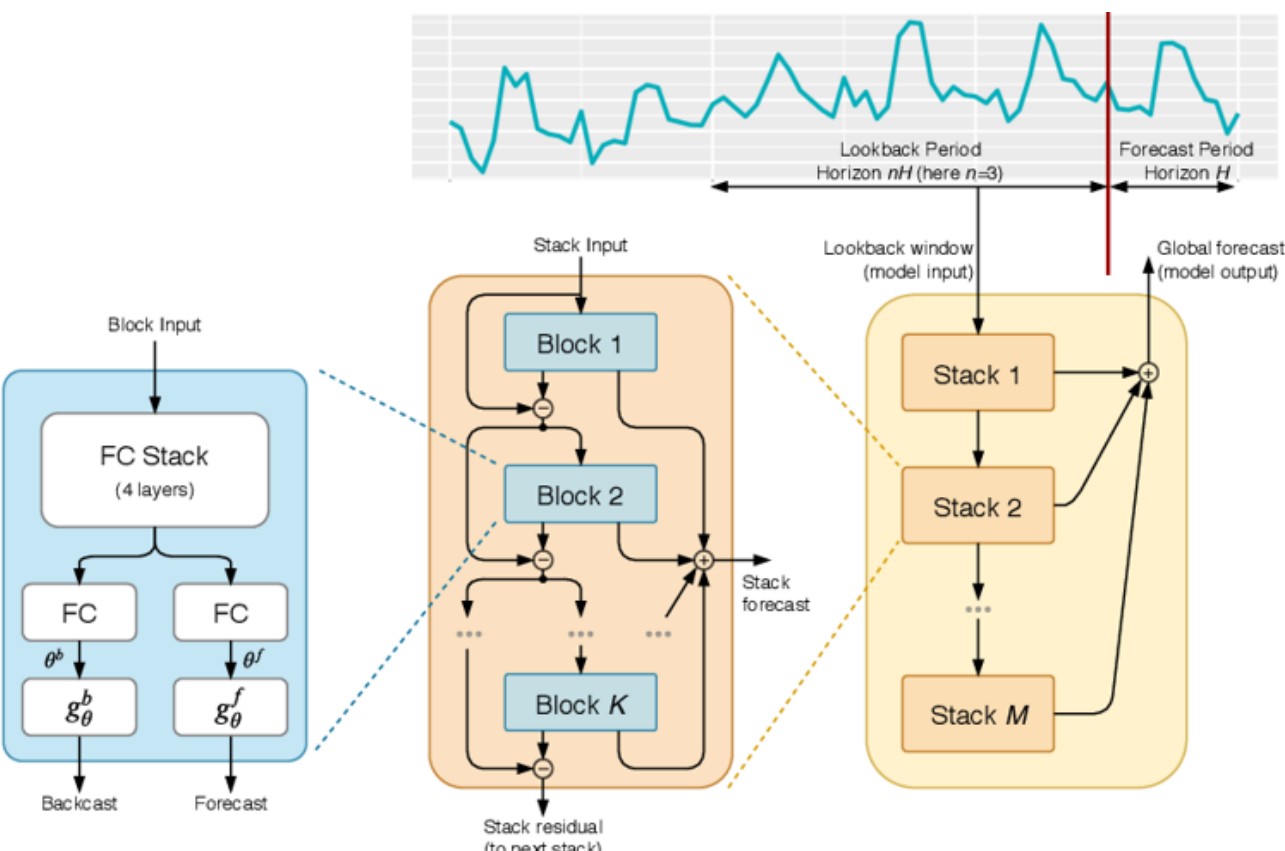

**Figure 2.** N-BEATS architecture (Reprinted from ref. [27]).

*4.4. Classification Method*

The labelling of the dataset is the initial stage in the creation of the classification model. Labelling helps us to understand the organisation and structure of the data. A contraction

is represented by the Boolean value of 1, whereas a non-contraction is represented by the Boolean value of 0. As one of the methods that is most frequently used in the literature, the K-means approach was selected as the labelling method. Due to this labelling, the construction of the classification model is simpler. In order to make an appropriate choice of classification technique, the K-fold approach was used for the comparison between three commonly used methods in the literature: XG-Boost (XGB), K-Nearest Neighbour (KNN), and Support Vector Machine (SVM).

## 5. Results

### 5.1. Labelling and Classification Method Choice Results

By applying the K-means method to our dataset, we were able to obtain Figure 3 below. The reader should recall that the data were normalised using the X_norm formula before K-means was applied.

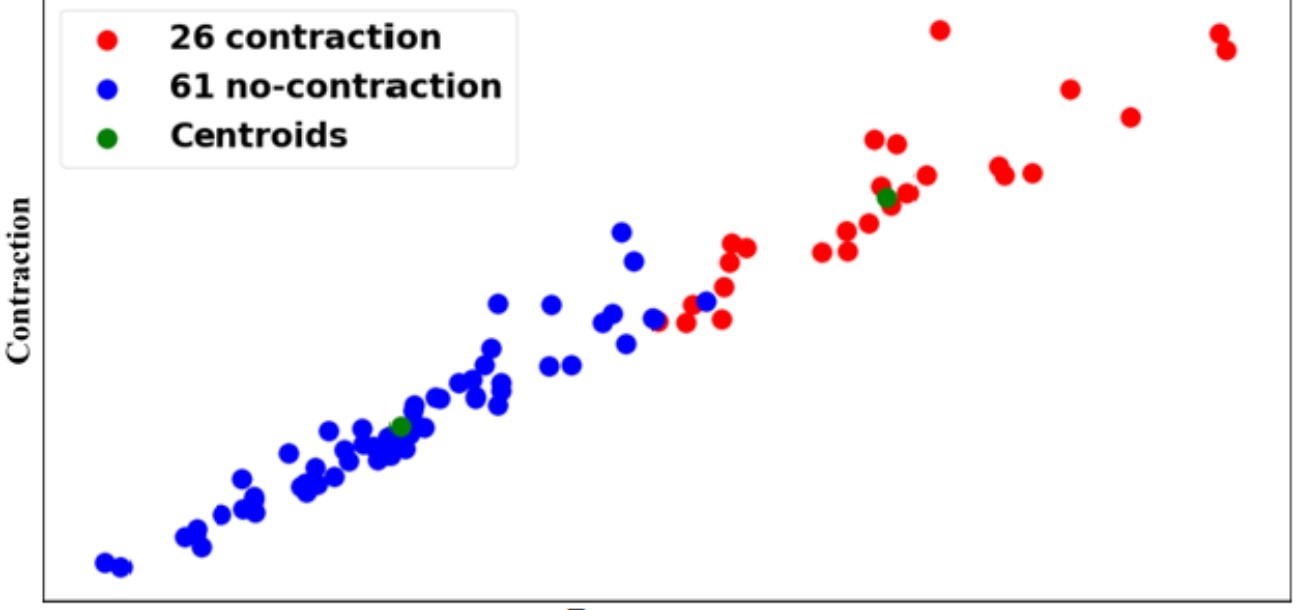

**Figure 3.** Application of K-means to the dataset.

Table 1 shows the scores of the different algorithms following their application to our dataset.

**Table 1.** Scores of the classification algorithms for different signals.

| Patient ID | Gestational Week | Accuracy | | | Recall | | | F1-Score | | |
|---|---|---|---|---|---|---|---|---|---|---|
| | | XGB | KNN | SVM | XGB | KNN | SVM | XGB | KNN | SVM |
| 1463 | 33 | 1.00 | 1.00 | 1.00 | 1.00 | 1.00 | 1.00 | 1.00 | 1.00 | 1.00 |
| 615 | 33 | 0.97 | 0.97 | 0.94 | 0.97 | 0.97 | 0.94 | 0.97 | 0.97 | 0.94 |
| 1745 | 33 | 1.00 | 1.00 | 0.97 | 1.00 | 1.00 | 0.96 | 1.00 | 1.00 | 0.97 |
| 1737 | 35 | 1.00 | 0.94 | 1.00 | 1.00 | 0.93 | 1.00 | 1.00 | 0.93 | 1.00 |
| **Average** | | **0.99** | **0.98** | **0.98** | **0.99** | **0.98** | **0.98** | **0.99** | **0.98** | **0.98** |

As shown in Table 1, XGB performed better than KNN and SVM. The XGB was therefore the classification algorithm of choice in this study. The confusion matrix for a given signal using XGB performed well, as it led to an RMSE = 0.00, precision = 1.00, and recall = 1.00.

### 5.2. Forecasting Results

In this sub-section, the results of the application of the proposed N-BEATS model to our dataset are presented. To train the model, we defined an input that corresponded to twice the forecast window.

The metric used was the symmetric mean absolute percentage error (*SMAPE*) [27], which rescales the error by the mean between the forecast and the exact values:

$$SMAPE = \frac{200}{H} \sum_{i=1}^{H} \frac{|y_{T+i} - \hat{y}_{T+i}|}{|y_{T+i}| + |\hat{y}_{T+i}|} \tag{1}$$

with $H$ as the forecasting horizon;
$y_{T+i}$: the exact or current value;
$\hat{y}_{T+i}$: the forecast value.

We started with the prediction horizon of at least 75 s. Table 2, below, shows the *SMAPE* values for horizons greater than or equal to 75 s.

**Table 2.** Dataset forecasting *SMAPE* and loss values for different horizons.

| Method | Epoch | Horizon | *SMAPE* |
|---|---|---|---|
| | 02 | 1500 (75 s) | 146.8 |
| | 02 | 1800 (90 s) | 146.4 |
| | 02 | 2400 (120 s) | 149.8 |
| | 02 | 3000 (150 s) | 147.5 |
| | 03 | 1500 (75 s) | 147.1 |
| | 03 | 1800 (90 s) | 147.6 |
| N-BEATS | 03 | 2400 (120 s) | 147.3 |
| | 03 | 3000 (150 s) | 146.7 |
| | 05 | 1500 (75 s) | 154.9 |
| | 05 | 1800 (90 s) | 156.2 |
| | 05 | 2400 (120 s) | 147.5 |
| | 05 | 3000 (150 s) | 144.3 |

As we can see, the best signal forecasting result was obtained for the horizon equal to 90 s. We can observe that we achieved the optimal performance within 3 epochs. Figure 4, below, shows the best forecasting result after applying the N-BEATS method.

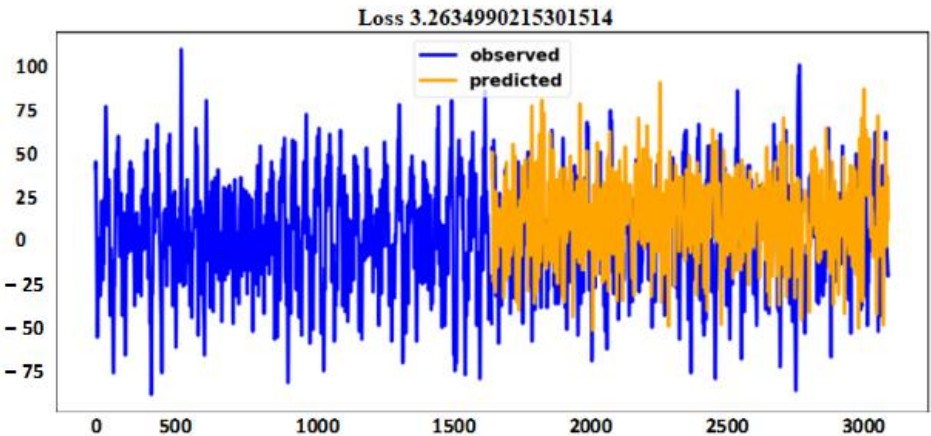

**Figure 4.** Application of the N-BEATS forecasting method.

### 5.3. Classification Results

The EHG signal is obtained after the forecast is sent to the classification model. As its output, we obtain the number of contractions and non-contractions. Let $c$ be the number

of contractions and $\hat{c}$ be the number of non-contractions. We calculate the probability of contractions as:

$$p = \frac{c}{c + \hat{c}} \tag{2}$$

Depending on the value of $p$, decisions are made:

- If $p < 0.5$, then $c < \hat{c}$, which means that we obtain a small number of contractions during this period;
- If $p \geq 0.5$, then $c \geq \hat{c}$, which means that we have a large number of contractions during this period.

In the first case, the woman is less likely to go into labour, whereas in the second case, the probability is high, because the more contractions the woman has, the closer she is to labour.

According to our results, we were able to detect the different types of contractions of pregnant women. Moreover, we were able to predict the activity of the contractions in order to determine the period of labour of the respective woman. Our results, when compared to the other classification methods in the literature using features in two or more domains, gave the following Table 3.

**Table 3.** EHG signal classification methods with features in two or more domains.

| Authors | Features | Method/Accuracy (%) |
|---|---|---|
| [10] | Fractal Dimension ($F_D^x$), Fuzzy Entropy ($E_F^x$), Interquartile Range ($I_{QR}^x$), Mean Absolute, Deviation ($AD_M^x$), Mean Energy ($\Omega_M^x$), Mean Teager–Kaiser Energy ($\Omega_{MTK}^x$), Sample Entropy ($E_S^x$), Standard Deviation ($D_S^x$) | Support Vector Machine using Radial Basis Function/96.25 |
| [32] | Wavelet Transform, Sample Entropy | Stacked Spark Autoencoder/90.00 |
| [20] | 31 features from the Time Domain, Frequency Domain, Time–Frequency Domain, and Non-Linear Analysis | Random Forest/93.00 |
| [11] | PKF, MNF, MF, Pregnancy Gestational Age, Pregnant Woman's Age, Parity | ANN/98.00 |
| **Our study** | RMS, MF, TM5, SM3, MYOP, ApEn, Tr, MAV2, TM3, SE, VCF, MNF, LOG, ZC, MAV1 | **XG-Boost/99.00** |

From Table A1, we can notice that Random Forest, SVM, and ANN are the best performing EHG signal classification methods in the literature. However, in the class of methods using features in two or more domains, our method remains the best performing, with an accuracy of 99%. It is clear that the use of a large number of parameters from different domains improves the performance of classification algorithms. Even though frequency and non-linear parameters can detect and predict preterm births, in the context of a full-term pregnancy, parameters from other domains should be combined with them, especially because it has been shown that the characteristics of EHG signals depend on anthropometric and pregnancy variables. Those that correspond to the signals of one pregnant woman do not always correspond to those of another. This justifies the importance of the step of the automatic selection of classification parameters in each case, which we proposed here.

It is also the only method that provides the classification of uterine contractions after the forecasting of EHG signal over a given horizon. This work provides a reference for the field of uterine labour contraction prediction techniques. Even if it does not allow for the forecasting of the EHG signal over a large horizon (in terms of hours), the objective of this work was to show that this is possible, as in the case of other time series. Future work will develop better forecasting models to enable the real-time prediction of labour in near-term pregnant women.

## 6. Discussion

There are very few examples in the literature of deep learning methods' application to EHG signals for labour prediction [16,17,34,42], with the vast majority of the work being based on machine learning. To the best of our knowledge, no study has examined EHG signal forecasting. The study that made a comparison between deep learning and machine learning methods showed that the Random Forest (machine learning method) gave better results than the ANN [16] in discrete classification. This work also demonstrates that the limitations of previous work can be found in the use of a single database, limitations that the authors attempted to overcome by using five databases.

In contrast to previous works, our work proposed a novel technique for predicting labour through EHG signal forecasting. The forecasting of physiological signals and, in particular, EHG signals is still at the immature stage, and few studies have addressed it. According to the literature, very few deep learning methods have been developed in this context and adapted to the healthcare sector. Moreover, of the most widely used methods in the healthcare sector, none has been used in the field of maternal and child health or for EHG signal forecasting. They are far more frequently used for disease prediction, in general, and cardiovascular disease in particular [20]. Therefore, our work suggested the N-BEATS as the LSTM architecture technique for forecasting EHG signals. However, the prediction time is constrained by the EHG signal's time interval and sampling frequency. An increase in the recording time of EHG signals can rectify this situation and extend the forecast to a much earlier prediction horizon.

Another concept that emerges from this work is the use of the dimensionality reduction method through principal component analysis. Beyond this first role, for which this technique is most frequently used throughout the literature [29,34], we noticed that it can be used as a technique for objective feature selection (among the 30 most frequently used features) so as to be exploited for each patient's EHG signal characterization. The authors often define a fixed number of parameters to be subjected to PCA [29] or seek to reduce the whole dataset after applying a method to it [34]. In contrast to existing works, we calculated the 30 most frequently used features (all domains) from the literature, which we subjected to PCA to extract the 15 representative principal components (all domains) of the signal being considered. Indeed, when applying the method to other examples, the parameters do not appear in the same order from one woman to another and from one week of gestation to another. This corroborates the results found in [26], with the conclusion that the characterisation features of an EHG signal depend on anthropometric and pregnancy variables. We therefore recommend that principal component analysis be applied to all the domains' features prior to any characterisation of the EHG signal and, thus, of uterine contractions. Thus, the parameters that do not truly represent a good proportion of the signal information can be discarded, unless one aims to specifically study the signal with respect to these parameters. It should also be remembered that these results will vary according to the number of parameters that are submitted, our work being one of the few to use so many parameters at once.

This result, although providing a good starting point for EHG signal forecasting, is not yet optimal for the achievement of our objectives. The prediction horizon should be evaluated in terms of hours, which would be an interesting investigation. This would give the pregnant woman time to prepare for the hospital. This is not the case with the results of the current forecasting model. This may be due to the multifractal nature of the EHG signal and the non-linearity of the EHG signal, as in the case of crude oil price time series [21,22], which make it difficult to forecast. The subject has not been meaningfully addressed in the literature, although it could provide a realistic alternative for healthcare systems in low-income countries. Meanwhile, research has been initiated in this direction in several other areas, such as electricity load [47,48], traffic [49] and COVID-19 mortality [50] forecasting. Additionally, the results prove that the LSTM [49] and hybrid models [47,48,50] perform better, especially in multivariate forecasting cases.

Accurate and long-term forecasting could enable the early prediction of labour in pregnant women in these countries. Achieving such a goal can help to overcome the lack of an adequate healthcare system [1] and thus reduce the infant and maternal mortality rates. This is why this study, as the first in a series of others yet to come, examined this subject. The results of this first work, which focused on the most efficient method of the RNN architecture for this signal, are encouraging.

As one can see, this is a preliminary work that may be affected by problems due to possible data overfitting. To avoid these drawbacks and obtain a better forecasting horizon (e.g., 1 h), our future work will focus on different aspects. Other versions of this method, as well as other architectures [20], will be applied to the EHG signal. This will help us to determine whether there an improvement in the forecasting performance can be obtained using the same data size. Another approach will be the integration of the trend and multifractality parameters of the EHG signal in forecasting, as proposed in [21,22], with the best results in the literature currently based on crude oil price forecasting. The implementation of our own EHG signal acquisition system, with the possibility of larger data sizes, is also a solution approach whose feasibility should be investigated.

## 7. Conclusions

In this study, we prototyped the forecasting of EHG signals for the purpose of labour prediction, as previously performed from the perspective of categorization. The N-BEATS model, which is based on a long short-term memory architecture, was recommended as one of the best models currently available in the other domains. The findings indicated that EHG signal recordings longer than 30 min, with the appropriate sample rate, are required for improved forecasting, which can result in a considerably earlier prediction of the labour contractions of the pregnant woman. Uterine contraction classification, following EHG signal forecasting, has been performed with an accuracy of 99%. This is one of the best classification performance accuracies based on the use of a good number of features. We believe that this hybrid model can enable the clinical application of labour prediction techniques. Future research will determine the appropriate recording time and sampling frequency required to achieve a given prediction goal. The other challenge is to study the applicability of other models used in the health domain to EHG signals and their comparison to N-BEATS signals. Our work indicates the importance of building larger datasets with longer recording times for hourly predictions and forecasting in future work. Additionally, it also suggests the importance of ensuring an increase in the forecasting horizon for timely labour predictions, resulting in efficient labour management.

**Author Contributions:** Conceptualization, methodology, writing—original draft, writing—review and editing, T.R.J. and Z.T.; methodology, software, visualization, formal analysis, writing—original draft, G.H. and M.H.A.; supervision, project administration, funding acquisition, A.E.-T.; supervision, project administration, D.M.; supervision, validation, project administration, A.L.; supervision, validation, F.S.; paper and English language review, publication funding acquisition, S.M.P.; paper review and editing, M.S.H. All authors have read and agreed to the published version of the manuscript.

**Funding:** This research was funded by the Al-Khawarizmi Program to Support Research in Artificial Intelligence and its Applications: funded by the Ministry of National Education, Professional Training, Higher Education and Scientific Research (MENFPESRS-Morocco); the National Centre for Scientific and Technical Research (CNRST-Morocco) and the Digital Development Agency (ADD-Morocco), grant number APIAA-2019-ETTAHIR-AZIZ-EST-RABAT-UM5R. The APC was funded by the University of Warwick.

**Informed Consent Statement:** Not applicable.

**Data Availability Statement:** The data used in this research were taken from the Term-Preterm EHG (TPEHG) database, which is publicly accessible through the following link: https://archive.physionet.org/cgi-bin/ATM, accessed on 15 July 2020.

**Conflicts of Interest:** The authors declare no conflict of interest.

## Appendix A

**Table A1.** Summary of the 30 most frequently used parameters in the literature [51–56].

| N° | Acronym | Name | Formula |
|---|---|---|---|
| | | Linear features | |
| | | Time domain | |
| 1 | IEMG | Integrated EHG | $\sum_{i=1}^{N}|X_i|$ |
| 2 | LOG | Log detector | $e^{\frac{1}{N}\sum_{i=1}^{N}\log(|X_i|)}$ |
| 3 | MAV | Mean absolute value | $\frac{1}{N}\sum_{i=1}^{N}|X_i|$ |
| 4 | WL | Wavelength | $\sum_{i=1}^{N-1}|X_{i+1}-X_i|$ |
| 5 | MAV1 | First modified mean absolute value | $\frac{1}{N}\sum_{i=1}^{N}W_i|X_i|$ <br> $W_i = \begin{cases} 1, & 0.25N \leq i \leq 0.75N \\ 0.5, & else \end{cases}$ |
| 6 | MAV2 | Second modified mean absolute value | $\frac{1}{N}\sum_{i=1}^{N}W_i|X_i|$ <br> $W_i = \begin{cases} 1, & 0.25N \leq i \leq 0.75N \\ \frac{4i}{N}, & i < 0.25N \\ \frac{4(i-N)}{N}, & else \end{cases}$ |
| 7 | AAC | Average amplitude change | $\frac{1}{N}\sum_{i=1}^{N-1}|X_{i+1}-X_i|$ |
| 8 | DASDV | Difference absolute standard deviation value | $\sqrt{\frac{1}{N-1}\sum_{i=1}^{N-1}(X_{i+1}-X_i)^2}$ |
| 9 | SSI | Simple square integral | $\sum_{i=1}^{N}X_i^2$ |
| 10 | MYOP | Myopulse percentage | $\frac{1}{N}\sum_{i=1}^{N}f(X_i)$ <br> $f(x) = \begin{cases} 1, & x \geq s \\ 0, & else \end{cases}$ |
| 11 | RMS | Root mean square | $\sqrt{\frac{1}{N}\sum_{i=1}^{N}X_i^2}$ |
| 12 | ZC | Zero crossing | $\sum_{i=1}^{N-1}sgn(X_i * X_{i+1}) * |X_i - X_{i+1}|$ <br> $sgn(x) = \begin{cases} 1, & x \geq s \\ 0, & else \end{cases}$ |
| 13 | VAR | EHG variance | $\frac{1}{N-1}\sum_{i=1}^{N}X_i^2$ |
| 14 | WAMP | Wilson amplitude | $\sum_{i=1}^{N-1}f(|X_i - X_i + 1|)$ <br> $f(x) = \begin{cases} 1, & x \geq s \\ 0, & else \end{cases}$ |
| 15 | TM3, TM4, TM5 | 3rd, 4th, 5th time moment absolute value | $\left|\frac{1}{N}\sum_{i=1}^{N}X_i^p\right|$ <br> $p = 3, 4, 5$ |
| 18 | SSC | Slope scale change | $\sum_{i=2}^{N-1}f[(X_i - X_{i-1}) * (X_i - X_{i+1})]$ <br> $f(x) = \begin{cases} 1, & x \geq s \\ 0, & else \end{cases}$ |

**Table A1.** *Cont.*

| N° | Acronym | Name | Formula |
|----|---------|------|---------|
| | | *Frequency domain* | |
| 19 | MNF | Mean frequency | $\frac{\sum_{j=1}^{M} f_j P_j}{\sum_{j=1}^{M} P_j}$ |
| 20 | MF | Median frequency | $\sum_{j=1}^{MF} P_j = \sum_{j=MF}^{M} P_j = \frac{1}{2} \sum_{j=1}^{M} P_j$ |
| 21 | MNP | Mean power | $\frac{\sum_{j=1}^{M} P_j}{M}$ |
| 22 | PKF | Peak frequency | $F\left(\max\left(P_j\right)\right)$ |
| 23 | TTP | Total power | $\sum_{j=1}^{M} P_j = SM0$ |
| 24 | SM1, SM2, SM3 | 1st, 2nd, 3rd spectral moment | $\sum_{j=1}^{M} P_j f_j^p$ <br> $p = 1,\ 2,\ 3$ |
| 27 | VCF | Variance of centre frequency | $\frac{SM2}{SM0} - \left(\frac{SM1}{SM0}\right)^2$ |
| | | **Nonlinear features (time domain)** | |
| 28 | ApEn | Approximate entropy | $A_p E_n(m,\ r,\ \tau,\ N) = \Phi^m(r) - \Phi^{m+1}(r)$ <br> $\Phi^m(r) = \frac{1}{N-(m-1)\tau} \sum_{i=1}^{N-(m-1)\tau} \log C_i^m(r)$ |
| 29 | SampEn (SE) | Sample entropy | $SE_{(m,r)}(r) = \begin{cases} -\log\left(\frac{C_m}{C_{m-1}}\right) : C_m \neq 0 \wedge C_{m-1} \neq 0 \\ -\log\left(\frac{N-m}{N-m-1}\right) : \quad C_m = 0 \wedge C_{m-1} = 0 \end{cases}$ |
| 30 | Tr | Time reversibility | $T_R(\tau) = \frac{1}{N-\tau} \sum_{n=\tau+1}^{N} (X_n - X_{n-\tau})^2$ |

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
