# Peer review of "N-Beats as an EHG Signal Forecasting Method for Labour Prediction in Full Term Pregnancy"

_electronics, doi:10.3390/electronics11223739_

Round 1

Reviewer 1 Report

The authors investigated a novel Neural Basis Expansion Analysis for Interpretable Time Series (N-BEATS) based model which predicts labour based on EHG forecasting and contraction classification over a given time horizon.

They found that Neural Basis Expansion Analysis for Interpretable Time Series (N-BEATS) forecasting can forecast EHG signals but the error increases with the number of epochs. Similarly, the forecasting signal's duration is determined by the length of the recordings. They then deployed XG-Boost which achieved the classification accuracy of 99 percent,  

Overall, this is an interesting study. I have no significant comments to this article. The following suggestions may help to further improve the manuscript:

1) Introduction. This section is quite long. Could it be a bit shortened?

2) Figure 2: Parts of Figure 2 are pretty small. Could you enlarge these?

3) Discussion. This section could be extended and the main aspect of your findings should be compared to similar studies in the current literature.

Author Response

Reviewer 1

Thank you for your precious feedback and comments. Please find highlighted in the text the relevant changes.

introduction:

This section is quite long. Could it be a bit shortened?

Many thanks for your valuable comment. As per advice, we reduced the introduction from 6 paragraphs to 4. Specifically the 4 previous last paragraphs have been transformed in 2 paragraphs.

Figure 2: Parts of Figure 2 are pretty small. Could you enlarge these?

Many thanks for your valuable comment. As per advice, small parts of figure 2 have been enlarged.

Discussion. This section could be extended and the main aspect of your findings should be compared to similar studies in the current literature.

Many thanks for your valuable comment. As per advice, the discussion has been extended and the main aspect of our findings has been compared to similar studies in the literature. Studies as (Almeida et al., 2022), (Karasu et al., 2020), (Karasu et Altan, 2022), (Ayub et al., 2020), (Shen et al, 2021), (Abduljabbar et al., 2021) and (Mathonsi et van Zyl, 2022) have been used for this purpose.

Reviewer 2 Report

This paper described a Neural Basis Expansion Analysis for Interpretable Time Series (N-BEATS) based method to predict labour based on EHG forecasting and contraction classification. They used a public dataset. I have to reject the paper due to low quality and lack of scientific soundness. 

The paper's figure is too unclear and low quality. The classification task is done like a common kaggle contest. Maybe the author can consider submit it to kaggle. The improvement from 98% to 99% is not of significance. It's may be arised from a dataset split issue or an overfitted evaluation.

The DL method is not well presented. Also, the predicted and actual data is not even compared closely.  

Therefore, I must reject it for publication.

Author Response

Reviewer 2

Thank you for your precious feedback and comments. Please find highlighted in the text the relevant changes.

The paper's figure is too unclear and low quality. The classification task is done like a common kaggle contest. Maybe the author can consider submit it to kaggle. The improvement from 98% to 99% is not of significance. It's may be arised from a dataset split issue or an overfitted evaluation.

The DL method is not well presented. Also, the predicted and actual data is not even compared closely.  

Many thanks for your valuable comments. As per advice, figures’ quality and clarity have been improved.

Regarding the accuracy of our classification method, we are well aware that with the size of the data we used, our results are subject to discussions such as over-fitting or class imbalance. As far as the over-fitting problem is concerned, it is only experimental data of appropriate size that can help to remove this doubt. We are not yet at this stage in our research. On the other hand, the objective of this work is to forecast the signal on a given horizon and then proceed to classification on this forecasted signal. It was therefore a question of choosing a classification method with a good performance. It turned out that the chosen method offered a 1% improvement over other methods in the literature, hence its choice. If we get our experimental data and this method no longer offers the same performance we will decide. We would like to point out that we made this choice and obtained this result based on the data used.

Reviewer 3 Report

Thank you for submitting the manuscript to Electronics. The author developed the Neural Basis Expansion Analysis for Interpretable Time Series (N-BEATS) model for predicting labour on full term pregnancy. This includes the use of principal component analysis (PCA) to reduce dimensionality and feature selection. This paper constructs a deeper architecture based on long short-term memory (LSTM) to solve the problem of EHG signal prediction. Finally, the performance of classification algorithms is selected and compared. This model combines the prediction and classification methods of EHG signal, and has achieved good results. It is a meaningful experiment, but this paper still has some shortcomings:

1.       Figures 1, 3, and 4 could be clearer.

2.       The author column in Table III is too wide.

Author Response

Reviewer 3

Thank you for your precious feedback and comments. Please find highlighted in the text the relevant changes.

Figures 1, 3, and 4 could be clearer.

Many thaanks for your valuable comments. As per advice, the clarity of figures 1, 3, and 4 has been improved.

The author column in Table III is too wide

Many thanks for your valuable comments. As per advice, the author column in Table III has been reduced.

Round 2

Reviewer 2 Report

First, the paper's novelty and originality are low, since the author did not improve the adopted ML methods or contribute to the dataset. The improved performance is negligible and not of interest to readers.

Second, the DL method's scientific soundness is low. The error increases with the 30 number of epochs, which is not supposed to be like this. The DL's model might be too complex for the small dataset, and it induced the overfitting.

Third, the quality of the figures are still low. How much dpi are you using? If you use the Snipping tool to copy the work "N-BEATS: Neural basis expansion analysis for interpretable time series forecasting" as your figure 2, is there a copyright problem? The author should be serious about copyright!

Author Response

Many thanks for your valuable comments. As per advice, to solve figure 2 copyright problem we have added the reference of the document in which the figure was taken after the title. We hope that we have solved the problem raised by the reviewer.

In terms of scientific novelty, this work is the pilot study of the application of deep learning based methods to predict labour based on EHG signal. The main purpose of this study is to demonstrate how real-time contractions can be predicted under low resource settings. To the best of our knowledge, this is the first study to develop deep learning based models predicting real-time contractions based on EHG signals. Currently we are focussing on improving the model in order to improve prediction horizon.

We can assure that we don’t have data overfitting problems as our training set has 300 records with each recording length of 30 minutes. For the future studies, we opt to have the training sets with longer recording lengths for better prediction horizon.

Round 3

Reviewer 2 Report

The revision is almost invisible. Please reference a good SCI indexed paper,and then come back to submit.

Author Response

Many thanks for your valuable comments. As per advice, we add the following references:

Leman, H.; Marque, C.; Gondry, J. Use of electrohysterogram signal for characterization of contractions during pregnancy. IEEE Transactions on Biomedical Engineering. 1999; Volume 46, no 10, pp. 1222‑1229. doi: 10.1109/10.790499

Buhimschi, C.; Boyle, M. B.; Garfield, R. E. Electrical activity of human uterus during pregnancy as recorded from the abdominal surface. Obstet Gynecol. 1997; Volume 90, pp. 102‑111. doi: 10.1016/s0029-7844(97)83837-9

Alberola-Rubio, J.; Prats-Boluda, G.; Ye-Lin, Y.; Valero, J.; Perales, A.; Garcia-Casaso, J. Comparison of non-invasive electrohysterographic recording techniques for monitoring uterine dynamics. Med Eng Phys. Dec 2013; Volume 35, no 12, pp. 1736‑1743. doi: 10.1016/j.medengphys.2013.07.008

Alexandersson, A.; Steimgrimsdottir, T.; Terrien, J.; Marque, C.; Karlsson, B. The icelandic 16-electrode electrohysterogram database. Sci Data. Dec 2015; Volume 2, no 150017. doi: 10.1038/sdata.2015.17

Jager, F.; Libenšek, S.; Geršak, K. Characterization and automatic classification of preterm and term uterine records. PLoS ONE. Aug. 2018; vol. 13, no 8, pp. 1‑49. doi: 10.1371/journal.pone.0202125